# Detection and Identification for Void of Concrete Structure by Air-Coupled Impact-Echo Method

**DOI:** 10.3390/s23136018

**Published:** 2023-06-29

**Authors:** Jinghui Ju, Xiushu Tian, Weigang Zhao, Yong Yang

**Affiliations:** 1School of Mechanical Engineering, Shijiazhuang Tiedao University, Shijiazhuang 050043, China; 2202002002@student.stdu.edu.cn; 2School of Material Science and Engineering, Shijiazhuang Tiedao University, Shijiazhuang 050043, China; 3School of Safety Engineering and Emergency Management, Shijiazhuang Tiedao University, Shijiazhuang 050043, China; zhaowg@stdu.edu.cn (W.Z.); yangy@stdu.edu.cn (Y.Y.)

**Keywords:** NDT, void width, void depth, peak frequency

## Abstract

In the field of non-destructive testing (NDT) for concrete structures, the traditional air-coupled impact-echo technology often has the problems of complex operation and low efficiency. In order to solve these problems, this study uses Comsol software to establish a finite element model (FEM) of the concrete structure with different void sizes and obtains the variation rule of peak frequency. The recognition property of the concrete void based on peak frequency is proposed, which is explained and validated by relevant theory and experiments. The results show that compared with the depth of the void, the influence of the void width on the peak frequency increases significantly. When the void width is greater than 0.3 m, the peak frequency of the sound wave decreases with the increase in the width, and the change is obvious. This paper describes the applicability of concrete void depth less than 0.4 m for the air-coupled method and, when the concrete void depth is less than 0.4 m, the peak frequency can be used to effectively identify void widths greater than 0.3 m. The research results will be beneficial to void detection of concrete structures such as tunnel lining and pavements.

## 1. Introduction

Concrete structures have the characteristics of high strength, good ductility and convenient construction, so they are widely used in bridges, high-rise buildings, tunnel linings and other structures. However, due to the long-term effect of site environment and other factors, different kinds of void will be produced in the concrete, which affect its strength [1,2,3].

Researchers use a variety of NDT techniques to detect and identify concrete damage. Tian et al. [4] used the impact-echo method to detect and identify the void defect of a CA mortar layer in CRTS Ⅱ ballastless track and proposed the identification parameters. Yang et al. [5] used the method of Burg power spectrum to solve the void imaging problem of ballastless track. Zhang et al. [6] used the ultrasonic method to test the filling amount of concrete voids for a steel–concrete bridge and proposed the layout type of the testing device, the basis and criterion of data processing and the layout principle of measuring area and testing point. Liu et al. [7] put forward the identification method of the equivalent diameter and area and the corresponding identification formula, according to the basic principle of ultrasonic detection technology and the characteristics of concrete-filled steel tube damage. Ding et al. [8] used the large-scale simulation test of a concrete-filled steel tube (CFST) to establish the force–optical constitutive relationship and the quantitative algorithm of voids and cracks in the actual engineering, identifying and evaluating the void damage of concrete structures effectively. Chen et al. [9] constructed a CNN to classify and identify the location and size of voids in a CA mortar layer. Lesicki et al. [10] identified spectra of concrete sliding damage by the impact-echo method and found that microstructural changes such as crack formation and debonding between separate phases in concrete are responsible for changes in the nonlinear parameter which can be tracked using the NIRAS testing technique. Bodnar et al. [11] proved that the void damage of concrete can be identified under low energy by the infrared thermal mapping technique. Yang et al. [12] used ground-penetrating radar to detect the ballastless track of a high-speed railway, analyzed the difference between steel echo and void echo and identified the void damage.

The above NDT methods have some limitations, e.g., the signals all belong to contact acquisition, and the sensitivity of different signals needs to be further studied. In recent years, with the continuous development of acoustic theory, the air-coupled sensor, as a non-contact sensor that can accept the acoustic wave information in the air, has been gradually applied to the research on the damage identification problems of civil engineering structures and materials. According to the different characteristics of the tested object, the frequency response range of the acoustic sensor can be selected from 50 to 1000 kHz. Among them, as a common air-coupled sensor, a microphone’s frequency response range is usually between 0 and 35 kHz, so it can be used to pick up the low-frequency sound wave leaked into the air by the concrete structure. Due to its advantages such as light volume and low price, it has been applied to the field of NDT by many scholars. Zhu et al. [13] used a microphone to acquire the acoustic signal of a concrete structure and found that inexpensive microphones are very effective to locate structural damage. Oh et al. [14,15,16] used microphones to identify shallow damage of concrete structures and optimized the imaging scanning system of shallow damage of bridge panels. Sun et al. [17] analyzed the acoustic signals acquired by microphones and proved that a ball chain, as an excitation method, could identify the void damage of concrete. Zhang et al. [18] analyzed signals of a microphone and designed a damage identification system for concrete bridge panels. Liu [19] made a comparison of the signal of an accelerometer and the signal of a microphone and proved that the air-coupled method can be used to identify void damage in concrete–steel structures. Shin et al. [20] found that a dynamic microphone successfully captures impact-echo signals in a contactless manner without acoustic shielding. Near-surface delaminations in the concrete slab were clearly identified, for which the obtained results are equivalent to those results obtained with a high-sensitivity sensor. Kim et al. [21] made use of a microphone to pick up Rayleigh waves to identify and evaluate the surface damage of a concrete structure. Dou et al. [22] proved that the fundamental frequency increases with the void depth by modal analysis, which provides theoretical support for the air-coupled method. Peng et al. [23] used acoustic frequency to identify the concrete void area and location.

The above discussion does not include the application range of the air-coupled method in the field of concrete. This paper proposes the method of peak sound frequency to identify concrete voids, in which the voids are located by the threshold of frequency and the void area is estimated by the multi-point scanning method, and the applicability of this method is discussed. The research results of this paper will provide a theoretical basis and method support for detection in concrete structures.

## 2. Theory

### 2.1. Theory of Void Damage Model

The model of concrete with void damage is shown in Figure 1. In the figure, it is assumed that the void damage is a cuboid with length *a*, width *b* and depth *h*.

The vibration of the concrete structure with a void is constrained by the surrounding concrete, and the boundary condition can be similar to the elastic boundary [24,25,26,27,28]. Different from fixed boundaries, the boundary condition can produce a turning angle. Different from the simply supported boundary, the boundary condition is affected by bending moment. Such boundary conditions are complex and difficult to define directly.

According to vibration theory of thin plates, the formula for calculating the natural frequency of simply supported thin plates on four sides is shown as Formula (1).
(1)$$\omega_{mn}=\pi^{2}\sqrt{\frac{D}{\rho h}}\left[{\left(\frac{m}{a}\right)}^{2}+{\left(\frac{n}{b}\right)}^{2}\right]$$
where $D=\frac{Eh^{3}}{12(1-\upsilon^{3})}$ is the bending stiffness of the thin plate, ρ is the material density, h is the geometric thickness of the thin plate, a and b are the length and width of the plate. m and n represent the modal order of the rectangular simply supported plate in its direction. Mitchell et al. [27,28] defined the parameters ${∆}_{m}$ and ${∆}_{n}$ corresponding to the order *m* and *n* of the modes of thin plates.

Thus, the edge effect of the constrained thin plate is considered in detail.
(2)$$\Delta_{m}=\left(a/\lambda_{a}\right)-m,\Delta_{n}=\left(b/\lambda_{b}\right)-n$$
where a and b represent the boundary length of the rectangular thin plate, respectively,$\lambda_{a}$ and $\lambda_{b}$ represent the half-wave length of the thin plate in the longitudinal and transverse mode shapes, respectively. Therefore, the corresponding edge effect coefficients can be used to express Equation (1). The formulas for calculating the natural frequency of quadrilateral constrained rectangular plates considering the boundary effect are as follows:(3)$$\omega_{mn}=\pi^{2}\sqrt{\frac{D}{\rho h}}\left[{\left(\frac{m+\Delta_{m}}{a}\right)}^{2}+{\left(\frac{n+\Delta_{n}}{b}\right)}^{2}\right]$$

Mitchell and Hazell, through a series of experiments, concluded that the edge effect coefficient can be found as follows:(4)$$\Delta_{m}={\left({\left(na/mb\right)}^{2}+c\right)}^{-1},\Delta_{n}={\left({\left(mb/na\right)}^{2}+c\right)}^{-1}$$

In general, the value of experience coefficient c is set to 2.

The research results of Cheng et al. [29] show that when the depth of defects remains unchanged, the natural frequency of vibration of rectangular defects in concrete structures is mainly determined by the width. Let the four sides of the void have the same length, then *a = b*. Combined with the bending stiffness *D* of the plate, Equation (3) is further modified to be
(5)$$\omega_{mn}=\pi^{2}\sqrt{\frac{D}{\rho h}}\left[{\left(\frac{m+\Delta_{m}}{a}\right)}^{2}+{\left(\frac{n+\Delta_{n}}{a}\right)}^{2}\right]$$

Formula (5) can effectively solve the vibration problem of thin plates, but there is error for medium-thickness plates. Zhao [30] introduced parameter *β*, which could effectively solve the problem of medium-thickness plates.
(6)$$\beta =2.29{\left(\frac{h}{a}\right)}^{1.54}+1.06$$
where *β* is the expansion coefficient of the width, the range of the parameter *β* is 1~1.6. Then, Formula (5) is expressed as
(7)$$\omega_{mn}=\pi^{2}\sqrt{\frac{D}{\rho h}}\left[{\left(\frac{m+\Delta_{m}}{a\beta }\right)}^{2}+{\left(\frac{n+\Delta_{n}}{a\beta }\right)}^{2}\right]$$

Combined with the bending stiffness *D* of the plate, Equation (3) is further modified to be
(8)$$\omega_{mn}=\pi^{2}\frac{h}{a^{2}}\sqrt{\frac{E}{12\rho (1-\upsilon^{3})}}\left[{\left(\frac{m+\Delta_{m}}{\beta }\right)}^{2}+{\left(\frac{n+\Delta_{n}}{\beta }\right)}^{2}\right]$$

In analysis by Equation (8), the natural frequency of the concrete plate is proportional to $\frac{h}{a^{2}}$.

### 2.2. Acoustic Modal Theory

According to the principle of sound and vibration reciprocity, the formula is expressed as follows.
(9)$$\frac{p_{j}}{F_{i}}{|}_{q_{j}=0}=-\frac{x_{i}}{q_{j}}{|}_{F_{i}=0}$$

When the force excitation is applied to the point *i* of the structure, the sound pressure response will be generated at the point j, and the frequency response function is obtained in this process. When the volume source excitation is applied to the point j of the structure, the velocity response will be generated at the point *i*, and the frequency response function is also obtained in this process. The frequency response functions of the two points mentioned above have the same magnitude and opposite directions.

For a structure with multiple degrees of freedom, when excited at point i, the strain frequency response function of point j is expressed as “$H_{ij}^{\varepsilon }$”:(10)$$H_{ij}^{\varepsilon }=\sum_{r=1}^{n}\frac{1}{-\omega_{r}^{2}M_{r}+j\omega_{r}C_{r}+K_{r}}\cdot \left\{\varphi_{jr}\cdot \phi_{ir}\right\}$$
where $\omega_{r}$ is the natural frequency of the structure, r is the natural frequency of the structure, $M_{r},C_{r},K_{r}$ are the *r*-th modal participation factors of the structure, n is the number of modes, $\phi_{ir}$ is the *r*-th displacement mode of point i. Let r be the spatial observation point, $r_{s}$ the position of the vibrating element on the surface of the structure and R the distance between the two points.
(11)$$p_{i}(\omega )=\int_{s}\frac{j\omega \rho_{0}v_{n}}{2\pi R}e^{-jkR}dS$$

Then, the transfer function of the sound pressure is shown as Formula (12).
(12)$$H_{ij}^{p}(\omega )=\frac{p_{i}(\omega )}{F_{j}(\omega )}=\frac{\int_{s}\frac{j\omega \rho_{0}v_{n}}{2\pi R}}{F_{j}(\omega )}=\int_{s}\frac{j\omega \rho_{0}e^{-jkR}}{2\pi R}H_{nj}^{v}(\omega )dS$$
where $H_{ij}^{p}(\omega )$ is the sound pressure frequency response function obtained in the case of point i and point j.ω is the circular frequency of the sound wave, $\rho_{0}$ is the density of the fluid medium, v is the vibration velocity of the sound source, R is the linear distance between the sound source and the measuring point and k is the acoustic transfer constant. S is the area of the sound source perpendicular to the direction of sound wave propagation. $H_{nj}^{v}(\omega )$ is the frequency response function of the vibration velocity of each vibration element in the structure when excited at n.

The influence of other modes at this frequency is ignored in *r*-th mode, then Formula (12) is expressed as
(13)$$[H_{ij}^{p}(\omega ){]}_{r}=\frac{\psi_{jr}}{m_{r}[(\omega_{r}^{2}-\omega^{2})+2j\xi_{r}\omega_{r}\omega ]}\int_{s}\frac{-\omega^{2}\rho_{0}e^{-jkR}}{2\pi R}\psi_{nr}dS$$

Formula (13) is the expression of the sound frequency response function of the *r*-th mode, which is similar to the expression of the strain mode.

### 2.3. Quantitative Index

In this paper, quantitative indexes k are proposed, where k is the rate of void area identification. It is used to evaluate the accuracy of void area identification.
(14)$$k_{2}=\frac{\left|S^{\prime }-S\right|}{S}\times 100\%$$
where S represents the actual void area. $S^{\prime }$ represents the void identification area.

## 3. Finite Element Model (FEM) for Concrete Void

### 3.1. Parameters of FEM

In order to study the acoustic characteristics of voids in the numerical model, the 3D numerical model of a 100 cm × 100 cm × 50 cm concrete slab with void damage is established by Comsol software, and the width of the void is set as *C* and the depth is set as *h*, as shown in Figure 2a. The acoustic structure interaction unit in the acoustic module is selected for the model, the boundary condition of the air domain is set as the plane wave radiation, the boundary condition of the contact between the concrete domain and the air domain is set as the acoustic–structure interaction of multiple physical fields, the bottom of the concrete model is set as a fixed constraint. The maximum grid in the air domain is set as 10 mm, and the maximum grid in the concrete domain is set as 60 mm. The mode of excitation force is set as point load. The peak value of impact force is set as 3000 N, as shown in Figure 2b.

Table 1 shows the material parameters of the FEM:

### 3.2. Sound Field Analysis of the FEM

#### 3.2.1. Analysis for the Influence of Width on Sound Field

According to the FEM, the central point of the concrete structure is excited, h = 10 cm is kept unchanged and C is set at 0 cm, 10 cm, 20 cm, 30 cm, 35 cm, 40 cm, 50 cm and 60 cm. The sound field images are captured at 2 ms.

Through the analysis of the FEM, it is found that when the width void C < 0.3 m, the sound pressure isosurface changes significantly, but it is not clear enough. As the width of the concrete void is relatively small, it presents the characteristics of a thick plate, and the acoustic modes are relatively complex. When the void width C ≥ 0.3 m, it can be clearly observed that the sound pressure isosurface is similar to a round cake above the concrete structure. As the width of the void increases, the isosurface decreases and the sound pressure value increases.

#### 3.2.2. Analysis for the Influence of Depth on Sound Field

According to the FEM, the central point of the concrete structure is excited, c = 50 cm is kept unchanged and *h* is set at 10 cm, 20 cm, 30 cm, 40 cm. The sound field images are captured at 2 ms.

Figure 3 shows that when the void depth increases, the sound pressure isosurface is almost constant, and the sound pressure value of the void is decreased gradually. Compared with Figure 4, it is found that the influence of the void width on the frequency is much greater than the void depth.

### 3.3. Frequency-Domain Analysis of the FEM

#### 3.3.1. The Influence of Different Excitation Positions

The void width is set as 45 cm × 45 cm and the depth set as 10 cm in the FEM. The microphone is placed 20 cm directly above the excitation point. The excitation points are set as A, B, C, D, E, F, G on the surface of the concrete structure as shown in Figure 3a, in which the red line is set as the position of the void, the point A is the geometric center for the upper surface of the void and the distance between 7 points is equal. The acoustic signals generated by different excitation points A, B, C, D, E, F and G are analyzed in the frequency domain as shown in Figure 5.

Figure 5a shows that the peak frequencies of the excitation points A, B, C, D and E within the range of the void are 1333 Hz and 1354 Hz, which mainly come from the vibration of the plate above the void. The peak frequencies of the excitation points F and G outside the range of the void is 2375 Hz, which mainly come from the vibration of the concrete structure.

#### 3.3.2. The Influence of Different Void Sizes

The microphone is positioned 20 cm directly above the void in the FEM. Data were collected for frequency-domain analysis, as shown in the Figure 6.

From Figure 6a–d, it can be found that when the void width C < 0.20 m, the peak frequencies of sound waves are equal. When the void width C ≥ 0.30 m, the peak frequency of the sound wave decreases as the void width increases.

Figure 6e shows that when the void width C < 0.3 m, the peak frequencies of different depths are completely coincident. It indicates that the peak frequency cannot effectively identify the concrete void with a width less than 0.3 m. When the void width C ≥ 0.3 m, the peak frequency decreases as the void width increases, which indicates that the peak frequency represents the mode of vibration of the plate.

In analysis based on Formula (8), when the void width C < 0.30 m, the void depth has little influence on the frequency of the sound wave, and there is no obvious correlation between the frequency and the depth h in the range. When the void width C > 0.30 m, the influence of void depth increases gradually. Through observation, it is found that the influence of the void width on the frequency is greater than the void depth, which confirms that the frequency is proportional to $\frac{h}{a^{2}}$, that is, the frequency is inversely proportional to $a^{2}$ and proportional to h. The result is consistent with Formula (8).

## 4. Experimental Verification

### 4.1. Experiment

In order to test verify the acoustic properties of concrete structures, three concrete models were made, namely Model A, Model B and Model C, as shown in Figure 7. The size of the concrete structures is 100 cm × 100 cm × 50 cm, the void is replaced by an air bag, and their sizes were as follows: Model A: 20 cm × 45 cm, *h* = 10 cm; Model B: 35 cm × 35 cm, *h* = 6 cm; Model C: 45 cm × 45 cm, *h* = 10 cm. Grid lines are shown on the surface of the models, the distance between the grid lines is 10 cm, and the grid lines show the actual position of the void. The experiment equipment mainly includes a microphone, Sirius high-speed acquisition instrument, electromagnetic hammer and computer. It is worth noting that the electromagnetic hammer can produce constant force (Figure 7).

### 4.2. Frequency-Domain Analysis of Experiment

The microphone is placed 20 cm directly above the excitation point. The acoustic signal are picked up and the frequency domain signals are analyzed.

Figure 8 shows that the peak frequencies of a, b and c in the model are almost identical, and the peak frequency is about 2440 Hz. It represents the overall vibration of the concrete structure, which proves that the microphone cannot effectively identify the void of width C = 0.2 m, which is consistent with the numerical simulation results.

Figure 9 shows that the peak frequency of a1 and b1 in the model is 1860 Hz, which can be used as a feature point for void identification. When the point c1 is hit, the concrete mode is complex, so it cannot be used as an effective node for judging the void.

Figure 10 shows that when the point a2, in the center of void, is hit, an obvious unimodal shape appears and the peak frequency was 1446 Hz. When point b2 near the edge of the void is hit, two peak frequencies appear in the frequency domain, and the first-order mode frequency value is 1446 Hz, which was generated by the vibration of the plate above the void. When hitting point c2 and point d2, the peak frequency is 2440 Hz, indicating that the frequency is generated by the vibration of the concrete structure.

## 5. Result Analysis and the Identification Method

### 5.1. Result Analysis of Experiment and FEM

The fundamental frequency of the void in Model B and Model C is peak frequency. The theory, FEM and experiment are compared, refer to Table 2.

It is found that data of Model B are closer than that of Model C. This is because the width-to-depth ratio of the plate in Model C is smaller, belonging to the medium-thickness plate, while the width-to-depth ratio of the plate in Model B is larger. Model C is closer to the thin plate. Due to the influence of shear deformation and extrusion deformation, there is error in medium-thickness plate theory and FEM.

### 5.2. The Method and Effect of Identification

The identification method is designed as follows:

Step 1: Draw the grid lines, select three nodes without voids, collect acoustic signals with a microphone and calculate the average value of peak frequency $f_{1}$ as the identification threshold value.

Step 2: Collect acoustic signals of all grid nodes by microphone, record the peak frequency f of each node.

Step 3: Calculate Δf by $\Delta f=f-f_{1}$ and generate a grayscale map. The darker color is the range of the void.

Each grid intersection is tapped in turn to pick up the microphone sound pressure signal directly above the point. The values of Δf are calculated as shown in Table 3 and Table 4.

Table 3 and Table 4 show that the specific location and general shape of the void can be identified through peak sound frequency. When the excitation point is located at the junction of the void area and the non-void area, the accuracy of void identification will be affected.

Figure 11 show that the dark part of the image is the identified void range, and the red dotted line is the actual void range. The experiment proves that the acoustic method can effectively identify the void.

The parameter *k* is the rate of the void area identification calculated by Formula (14), where $S^{\prime }$ is the void identification area. S is the actual void identification area. k is calculated in Table 5.

Table 5 shows that the area recognition rate of Model C by peak frequency is higher than the area recognition rate of Model B, indicating that the void depth is smaller and the area recognition rate is larger.

## 6. Discussion

This paper mainly focuses on the application range of the peak frequency method to detect concrete voids. When the plate above the void is a thin plate, the coefficient β in Formula (8) is approximately equal to 1, and the value of frequency *f* is determined by $\frac{h}{a^{2}}$. When the plate above the void is a medium plate, the coefficient β will increase. Formula (8) can be used to calculate the depth of the concrete void, and there will be some errors in a medium plate (thin plate and medium plate are determined by the width to thickness ratio). This paper mainly discusses the application range of acoustic peak frequency for a medium plate. The next research work for the research team is to improve the recognition accuracy by a multi-parameter fusion method.

## 7. Conclusions

(1)In this paper, numerical simulation and experiments are described for a void depth of less than 0.4 m. The results show that, compared with the void depth, the influence of the width on peak frequency increases significantly. When the void width is greater than 0.30 m, the peak frequency decreases with the increase in void width, and the change is obvious.(2)It is found that the acoustic peak frequency can effectively judge a concrete void depth of less than 0.4 m by numerical simulation. The method of peak frequency can be used identify a void with a width greater than 0.3 m in a concrete structure.(3)The main engineering value of this study is that the threshold value can be used to quickly judge whether there is a void in a concrete structure through single-point excitation. When multipoint scanning is used, the void range can be quickly estimated.

## Figures and Tables

**Figure 1 sensors-23-06018-f001:**
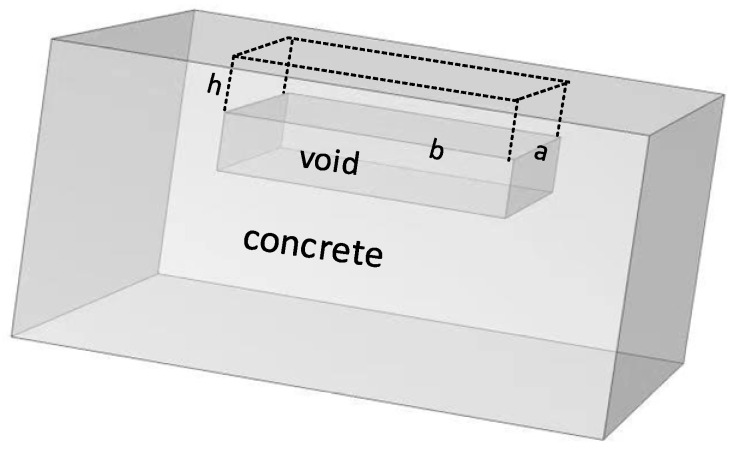
Model of concrete structure with void damage.

**Figure 2 sensors-23-06018-f002:**
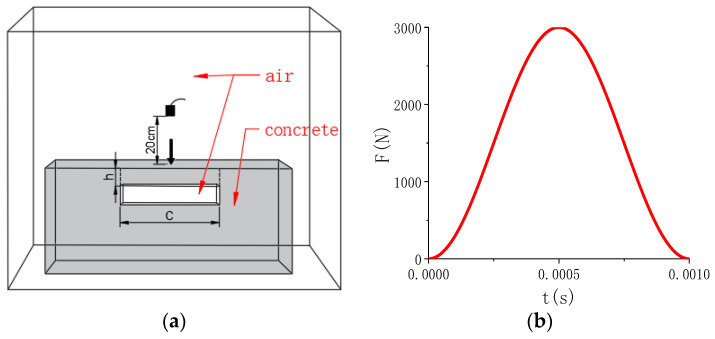
Finite element model (FEM) of concrete structure with void. (**a**) The FEM; (**b**) the impact force.

**Figure 3 sensors-23-06018-f003:**
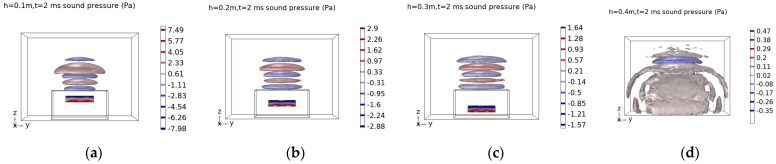
Sound field distribution map at t = 2 ms. (**a**) c = 50 cm, h = 10 cm; (**b**) c = 50 cm, h = 20 cm; (**c**) c = 50 cm, h = 30 cm; (**d**) c = 50 cm, h = 40 cm.

**Figure 4 sensors-23-06018-f004:**
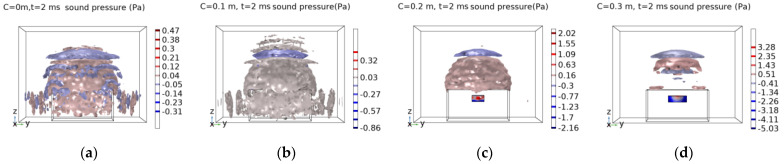
Sound field distribution map at t = 2 ms. (**a**) h = 10 cm, c = 0 cm; (**b**) h = 10 cm, c = 10 cm; (**c**) h = 10 cm, c = 20 cm; (**d**) h = 10 cm, c = 30 cm; (**e**) h = 10 cm, c = 35 cm; (**f**) h = 10 cm, c = 40 cm; (**g**) h = 10 cm, c = 50 cm; (**h**) h = 10 cm, c = 60 cm.

**Figure 5 sensors-23-06018-f005:**
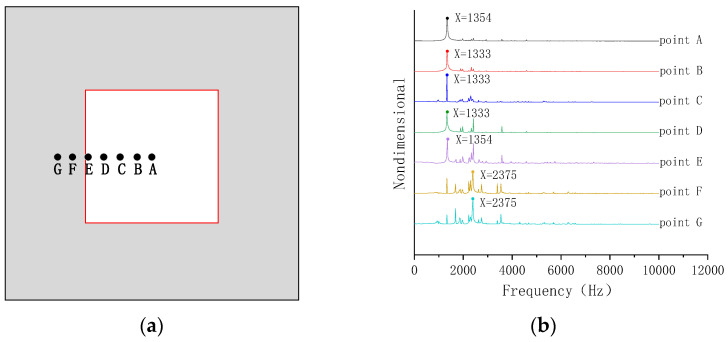
Frequency-domain analysis. (**a**) Layout of excitation point. (**b**) Frequency-domain waveform.

**Figure 6 sensors-23-06018-f006:**
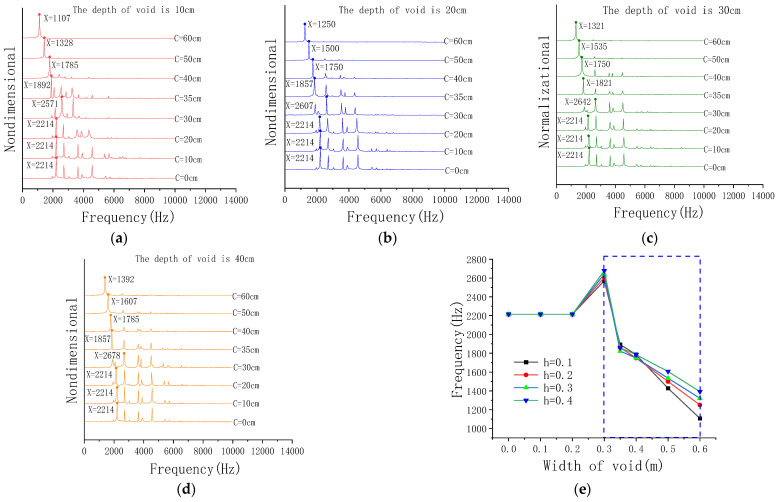
The void of concrete. (**a**) The void depth h = 0.1 m. (**b**) The void depth h = 0.2 m. (**c**) The void depth h = 0.3 m. (**d**) The void depth h = 0.4 m. (**e**) Line chart of peak frequency.

**Figure 7 sensors-23-06018-f007:**
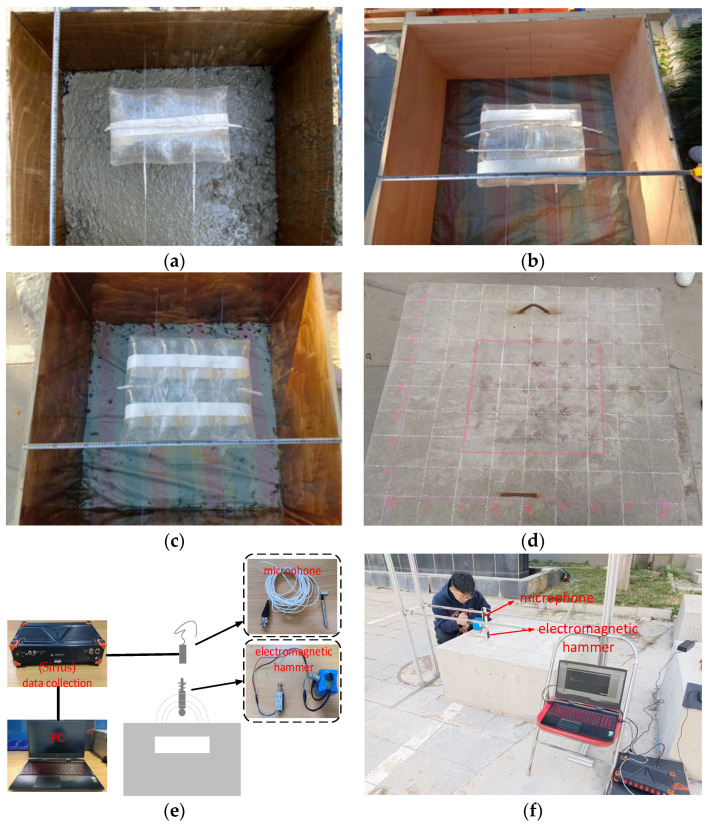
Experiment design (**a**) Model A: h = 10 cm, C = 20 cm; (**b**) Model B: h = 6 cm, C = 35 cm; (**c**) Model C: h = 10 cm, C = 45 cm; (**d**) grid line layout; (**e**) schematic plot; (**f**) experiment.

**Figure 8 sensors-23-06018-f008:**
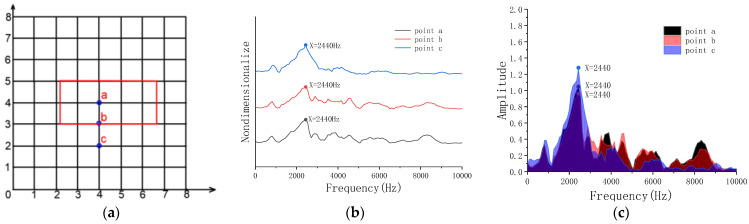
Frequency-domain analysis of Model A. (**a**) Grid lines; (**b**) peak frequency of excitation points a, b, c; (**c**) peak sound pressure of excitation points a, b, c.

**Figure 9 sensors-23-06018-f009:**
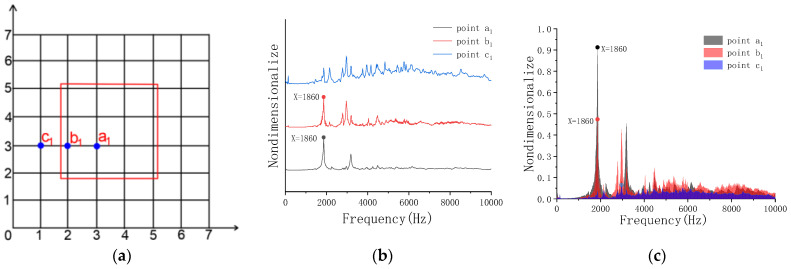
Frequency-domain analysis of Model B. (**a**) Grid lines; (**b**) peak frequency of excitation points a_1_, b_1_, c_1_; (**c**) peak sound pressure of excitation points a_1_, b_1_, c_1_.

**Figure 10 sensors-23-06018-f010:**
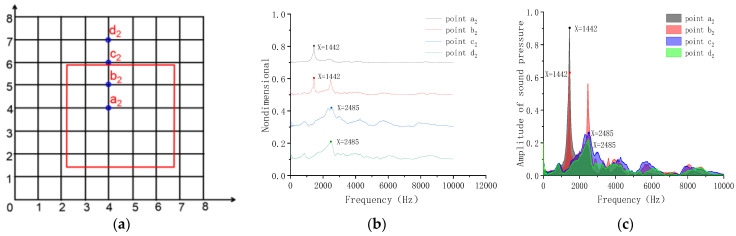
Frequency-domain analysis of Model C. (**a**) Grid lines; (**b**) peak frequency of excitation points a_2_, b_2_, c_2_ and d_2_; (**c**) peak sound pressure of excitation points a_2_, b_2_, c_2_ and d_2_.

**Figure 11 sensors-23-06018-f011:**
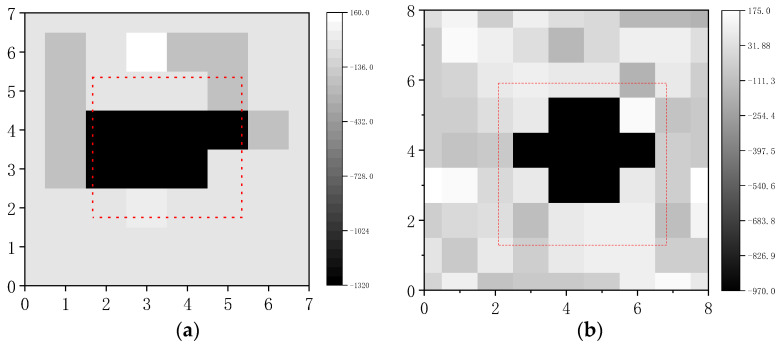
Image of peak frequency. (**a**) Grayscale image of Model B; (**b**) grayscale image of Model C.

**Table 1 sensors-23-06018-t001:** Material parameters of the model.

Materials	Velocity of Sound (m/s)	Density (kg/m^3^)	Elastic Modulus (Pa)	Poisson Ratio
Concrete	4000	2500	3.0 × 10^10^	0.2
Air	343	/	/	/

**Table 2 sensors-23-06018-t002:** **Comparison of** fundamental frequency.

*f*	Theory	FEM	Experiment
Model B	1862 Hz	1892 Hz	1860 Hz
Model C	1551 Hz	1354 Hz	1442 Hz

**Table 3 sensors-23-06018-t003:** Δ*f* of each node for Model B.

Δ*f* (Hz)	0	1	2	3	4	5	6	7
0	0	0	0	0	0	0	0	0
1	0	0	0	0	0	0	0	0
2	0	0	0	20	0	0	0	0
3	0	−220	−1320	−1320	−1320	0	0	0
4	0	−220	−1320	−1320	−1320	−1320	−200	0
5	0	−220	0	0	0	−220	0	0
6	0	−220	0	160	−1040	−220	0	0
7	0	0	0	0	0	0	0	0

**Table 4 sensors-23-06018-t004:** Δ*f* of each node for Model C.

Δ*f* (Hz)	0	1	2	3	4	5	6	7	8
0	−29	86	−100	−86	−86	−57	86	129	71
1	43	−71	57	−43	71	100	86	−57	−57
2	−43	0	14	−129	57	86	100	−114	114
3	157	129	0	71	−957	−970	57	−43	171
4	−57	−100	−86	−957	−957	−957	−957	−43	−86
5	−43	−57	14	71	−957	−957	143	−100	−71
6	−43	−29	71	86	57	71	−171	71	−57
7	−57	129	86	14	−157	−14	100	100	29
8	14	114	−57	100	29	−14	−157	−143	−176

**Table 5 sensors-23-06018-t005:** The rate of void area identification.

	Width of Void	Depth of Void	k
Model A	0.45 m	0.10 m	39.5%
Model B	0.35 m	0.06 m	57.1%

## Data Availability

The data in this study are available on request from the first author or corresponding author.

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
