# Peer review of "Detection and Identification for Void of Concrete Structure by Air-Coupled Impact-Echo Method"

_sensors, 2023, doi:10.3390/s23136018_

Round 1

Reviewer 1 Report

This article proposes a method for identifying concrete void damage
based on peak sound pressure and peak frequency.   In my opinion, the
research objectives are clear.   The content structure is reasonable.   The
experimental results seem acceptable and can support the feasibility of
this method.

However, several points should be addressed to improve the quality of
the document.

1.   In an abstract way, please use the correct tense to reduce the use of
complex and difficult sentence structures.   Suggest modifying the grammar
and presentation issues in the entire manuscript.

2.   Lack of literature research and insufficient depth.   In the introduction section, only a small amount of literature on concrete damage detection methods was briefly mentioned, and there was no relevant evaluation of these methods.

3.   The semantics and syntax errors of some sentences in the text are not
clear.   It is suggested to further modify the syntax and grammar of the
full text.

4.   The peak frequency method has a gap recognition rate of 54.3%, the
peak frequency method has a gap recognition rate of 39.5%

5.   Please explain in the text the applicability of the proposed method,
such as monitoring the size of the concrete cavity.

Has the author considered conducting comparative experiments with
different cavity sizes to further verify the feasibility of this method?

The semantic and grammatical errors of some sentences in this paper are unclear, so it is suggested to further modify the syntax and grammar of the whole paper.

Reviewer 2 Report

In this study, the authors aimed to demonstrate the use of threshold values of frequency and sound pressure for damage identification in concrete structures through a combination of theoretical models, finite element method (FEM) simulations, and experimental tests. While the presented approach has potential applications in non-destructive testing (NDT) of concrete structures, the manuscript suffers from several shortcomings that prevent its recommendation for publication. Detailed comments are provided below:

·       Lack of innovation: The study does not present significant novelty, as the use of threshold values of frequency and sound pressure for identifying damage in concrete structures is already a common practice in NDT. Additionally, multipoint scanning is widely employed in various NDT methods. Furthermore, the applicability of the peak frequency values appears to be limited to concrete structures with specific dimensions and may not be easily generalized to other cases.

·       Formatting issues: The manuscript's format is inconsistent and appears to deviate from the journal's requirements. For instance, the width of the references and some tables exceeds the manuscript's text width. The authors should verify whether this format is in accordance with the guidelines provided by Sensors.

·       Citation and referencing inconsistencies: The manuscript contains numerous errors and inconsistencies in the citation and referencing formats. The authors should carefully review and correct these issues before resubmission. For example, in line 65, "Zhu J et al." should be "Zhu et al."; in line 73, "Liu Meng" should be "Liu." Similar errors are present throughout the bibliography, where both full names and short names are used inconsistently. Only the last names should be shown in the manuscript, and the authors must ensure that the citation and referencing formats are consistent and follow the journal's guidelines.

The English quality is okay but can be improved. Some aspects can be improved are listed as below.

Prepositions: The authors seem to struggle with the correct usage of prepositions, which can lead to awkward or unclear phrasing. It's essential to ensure that the prepositions used fit the context and convey the intended meaning. Examples from the abstract include "of the void damage of concrete structure model" and "the identification method of the void damage."

Article usage: The authors should pay attention to the proper use of definite (the) and indefinite (a, an) articles. It is necessary to use articles correctly to provide clarity and maintain proper sentence structure. Examples from the abstract include "the void damage of a concrete structure model" and "the peak frequency of the concrete structure."

Consistency in terminology: It is crucial to maintain consistency in the terms used throughout the paper. In the abstract, the authors sometimes use "the range of the void" and other times "the void range" or "the range of voids." Using consistent terminology helps avoid confusion and ensures clear communication.

Reviewer 3 Report

The manuscript describes an approach of detection of voids in a concrete elements based on monitoring of two criteria, the frequency of peak response, and the amplitude of sound pressure. The approach is illustrated on a 1 m by 1 m slab with a void 45 cm by 45 cm, 10 cm below the surface. The presented numerical simulation and experimental results are pretty close.

There are two major questions the authors would need to provide explanations. First, the authors should provide the theoretical solution regarding the fundamental frequency of the presented model. The authors have written over two pages about the theoretical solution for the natural frequency of flexural oscillations of a rectangular thin plate, but have not compared the theoretical result with the numerical or experimental result. Second, the authors should speak about the thickness mode, predominantly used in IE testing, and whether it would be a more accurate way to detect the boundaries of the void. The thickness frequency for the 10 cm deep void (or delamination), is expected to be about 20 kHz. Have the authors evaluated the response in that frequency range? If not, why not.

Several minor comments:

1)      Line 19. The authors should not refer to specific frequencies. The frequency will depend on the size and depth of the void or delamination.

2)      Line 47. IT is unclear what the authors wanted to say with the phase angle of HMA.

3)      The authors should be consistent with citations in the main text. Write just last names.

4)      Line 79. Are the authors referring to Kim et al as reference 28 instead of 21? On the other hand, some other references are incorrectly numbered, like Mitchell, and Hazell. The authors should check all.

5)      I do not think that the authors should call void as void damage. Void would be more a result of poor construction, while damage would be more in a form of a crack.

6)      Line 150 – Mr, Cr, and Kr are probably mode participation factors, not matrices.

7)      Table 1 – Modulus should be 1010.

8)      Line 201 – Refer to Figure 3.

Only a few minor improvements should be made. No major issues.

Round 2

Reviewer 1 Report

The paper has been improved to a certain extent after the author's revision, but there are still some problems:

The applicability referred to is how much accuracy the method can detect, how wide a crack can be detected at a minimum and whether it has practical engineering use. In addition, the recognition accuracy is not high enough and whether it can be improved.

Minor editing of English language required.

Author Response

Response to Reviewer 1 Comments

The paper has been improved to a certain extent after the author's revision, but there are still some problems:

The applicability referred to is how much accuracy the method can detect, how wide a crack can be detected at a minimum and whether it has practical engineering use. In addition, the recognition accuracy is not high enough and whether it can be improved.

Response: This paper mainly discusses the application range of acoustic peak frequency for medium plate. The result show that the acoustic peak frequency can effectively judge the concrete void depth less than 0.4m. The method of peak frequency can be used identify the void with the width greater than 0.3m in the concrete structure. The research will be beneficial to the rapid identification of concrete voids in practical engineering such as Tunnel lining and pavement. The next research work for the research group is to improve the recognition accuracy by Multi-parameter fusion method. Thank you for kind advice.

Reviewer 2 Report

Still, the novelty is a big problem for this study and the authors have yet provided a straight forward answer to me. Nevertheless, given the substantial effort the authors have invested, I am tentatively willing to concede that the study meets the minimum standards for publication. I wish to underscore, however, that novelty should always be a primary focus in research. Additionally, I urge the authors to treat this response seriously. There are multiple instances where the word "many" is misspelled as "mang". In rectifying such errors and others that may be present, it would be beneficial for the authors to detail the modifications made in response to feedback, rather than merely acknowledging that these issues have been addressed.

I think that the quality of English can meet the minimum standard of publication.

Author Response

Response to Reviewer 2 Comments

Still, the novelty is a big problem for this study and the authors have yet provided a straight forward answer to me. Nevertheless, given the substantial effort the authors have invested, I am tentatively willing to concede that the study meets the minimum standards for publication. I wish to underscore, however, that novelty should always be a primary focus in research.

Response: Thank you very much for the valuable review comments provided for our articles. We attach great importance to your review conclusions,which will help improve the quality of our manuscripts.

Additionally, I urge the authors to treat this response seriously. There are multiple instances where the word "many" is misspelled as "mang". In rectifying such errors and others that may be present, it would be beneficial for the authors to detail the modifications made in response to feedback, rather than merely acknowledging that these issues have been addressed.

Response: We are very sorry for our negligence, where the word "many" is misspelled as "mang".we have carefully checked the article and corrected the relevant errors for example Table 2 is replaced by Table 3 in line 321,The word “less” is replaced by “greater” in lines 17 and 20. Thank you for kind advice.

Reviewer 3 Report

The authors have responded to the reviewer's comments and improved the manuscript. One main comment to which the authors did not respond is the discussion about the thickness mode, predominantly used in IE testing to measure the slab thickness or the depth of the void. The ability would also enable more accurate sizing of the void. The thickness frequency for the 10 cm deep void (or delamination), is expected to be about 20 kHz. Have the authors evaluated the response in that frequency range? The source used in numerical simulations with a peak frequency of about 1 kHz would be inappropriate for that. Have the authors explored using sources of much higher peak frequency, 20 kHz or so?

The authors should address the feasibility of detecting deeper voids from a practical point. The reviewer has serious doubts that the voids 0.3 m or 0.4 m deep would be detected and properly sized using air-coupled IE. From that point, the statement in lines 19 and 20 is a bit questionable and the second part is incorrect. It should say “cannot be used to identify….”

The author should also state how they chose the 20 cm height of the microphone. Seems to be too high, since a lot of pressure is lost.

Several other comments:

1)      The authors should specify in the FEM section what was the length of the void. Was it square?

2)      Line 224-226. A bit unclear statement regarding pressures. However, it demonstrates that it does not recognize differences.

3)      Line 259. I do not agree with the statement about the detectable void width. The detectability depends on the depth-to-width ratio. The authors should state in those terms.  

4)      Line 320. Should be Table 2 instead of 3.

5)      Tables 3 and 4. The authors should put units. It seems one is in Hz, the other in kHz.

6)      Line 357. The authors should define k.

7)      Conclusions 1 and 2. The authors should state that detection at those depths (03.m to 0.4 m) is feasible based on the numerical simulations. If they have not proven that experimentally, they should state so.

It is generally well written. No major issues.
